# Dinosaur bonebed amber from an original swamp forest soil

**Sergio Álvarez-Parra[1]\*, Ricardo Pérez-de la Fuente[2], Enrique Peñalver[3], Eduardo Barrón[3], Luis Alcalá[4], Jordi Pérez-Cano[1], Carles Martín-Closas[1], Khaled Trabelsi[5,6,7], Nieves Meléndez[8], Rafael López Del Valle[9], Rafael P Lozano[3], David Peris[1], Ana Rodrigo[3], Víctor Sarto i Monteys[10], Carlos A Bueno-Cebollada[3], César Menor-Salván[11,12], Marc Philippe[13], Alba Sánchez-García[14,15], Constanza Peña-Kairath[1], Antonio Arillo[16], Eduardo Espílez[4], Luis Mampel[4], Xavier Delclòs[1]**

[1]Departament de Dinàmica de la Terra i de l'Oceà and Institut de Recerca de la Biodiversitat (IRBio), Facultat de Ciències de la Terra, Universitat de Barcelona, c/ Martí i Franquès s/n, 08028, Barcelona, Spain; [2]Oxford University Museum of Natural History, Oxford, United Kingdom; [3]Museo Geominero (IGME, CSIC), c/ Ríos Rosas 23, Madrid, Spain; [4]Fundación Conjunto Paleontológico de Teruel-Dinópolis/Museo Aragonés de Paleontología, Av. Sagunto s/n, Teruel, Spain; [5]Université de Sfax, Faculté des Sciences de Sfax, Sfax, Tunisia; [6]Université de Tunis El Manar II, Faculté des Sciences de Tunis, LR18 ES07, Tunis, Tunisia; [7]Department of Geology, University of Vienna, UZA 2, Vienna, Austria; [8]Departamento de Geodinámica, Estratigrafía y Paleontología, Facultad de Ciencias Geológicas, Universidad Complutense de Madrid, Ciudad Universitaria, Madrid, Spain; [9]Museo de Ciencias Naturales de Álava, c/ Siervas de Jesús 24, 01001, Vitoria-Gasteiz, Spain; [10]Institut de Ciència i Tecnologia Ambientals (ICTA), Edifici Z – ICTA-ICP, Universitat Autònoma de Barcelona, Barcelona, Spain; [11]School of Chemistry and Biochemistry, Georgia Institute of Technology, Atlanta, United States; [12]Departamento de Biología de Sistemas/Instituto de Investigación Química "Andrés del Río" (IQAR), Universidad de Alcalá, 28805, Alcalá de Henares, Madrid, Spain; [13]Univ. Lyon, Université Claude Bernard Lyon 1, CNRS, ENTPE, UMR 5023 LEHNA, Villeurbanne, France; [14]Departament de Botànica i Geologia, Facultat de Ciències Biològiques, Universitat de València, c/ Dr. Moliner 50, Burjassot, Spain; [15]Division of Invertebrate Zoology, American Museum of Natural History, New York, United States; [16]Departamento de Biodiversidad, Ecología y Evolución, Facultad de Biología, Universidad Complutense de Madrid, c/ José Antonio Novais 12, Madrid, Spain

**\*For correspondence:**
sergio.alvarez-parra@ub.edu

**Competing interest:** The authors declare that no competing interests exist.

**Abstract** Dinosaur bonebeds with amber content, yet scarce, offer a superior wealth and quality of data on ancient terrestrial ecosystems. However, the preserved palaeodiversity and/or taphonomic characteristics of these exceptional localities had hitherto limited their palaeobiological potential. Here, we describe the amber from the Lower Cretaceous dinosaur bonebed of Ariño (Teruel, Spain) using a multidisciplinary approach. Amber is found in both a root layer with amber strictly in situ and a litter layer mainly composed of aerial pieces unusually rich in bioinclusions, encompassing 11 insect orders, arachnids, and a few plant and vertebrate remains, including a feather. Additional palaeontological data—charophytes, palynomorphs, ostracods— are provided. Ariño arguably represents the most prolific and palaeobiologically diverse locality in which fossil-iferous amber and a dinosaur bonebed have been found in association, and the only one known

where the vast majority of the palaeontological assemblage suffered no or low-grade pre-burial transport. This has unlocked unprecedentedly complete and reliable palaeoecological data out of two complementary windows of preservation—the bonebed and the amber—from the same site.

## Editor's evaluation

In an integrative way, the authors introduced an exceptional Konservat-Lagerstätte jointly preserving dinosaur remains and fossiliferous amber. Impressively, this is the first time that strictly in situ amber is reported, and the key claims of the manuscript are well supported by the paleontological and geochemical data. This manuscript will be of broad interest to scientists, including paleontologists, geobiologists, ecologists and geologists, as well as the public.

## Introduction

Localities preserving either vertebrate bonebeds or fossilised plant resin (amber) are among the most valuable sources of information on past terrestrial ecosystems (*Rogers et al., 2007*; *Seyfullah et al., 2018*). Yet, when a bonebed and fossilised resin are found jointly in the same site, and there is certainty that they originally belonged to the same biocoenosis, the potential for extracting and integrating palaeobiological data is barely unmatched in palaeontology. Although amber from the Cretaceous is often found together with other fossils such as plant and, more infrequently, vertebrate remains, fossiliferous amber associated with bonebeds including dinosaurs has been previously reported in only three occasions. Firstly, the lower Cenomanian (ca. 96–100.5 Ma) locality of Fouras/Bois Vert (Charente-Maritime, France) yielded diverse vertebrate remains, including about 50 dinosaur bone fragments, alongside plant macroremains, molluscs, and amber lumps, a few of which were fossiliferous (*Néraudeau et al., 2003*). From the latter, ~110 bioinclusions belonging to arachnids, springtails and, at least four insect orders have been reported, including several species described (*Perrichot et al., 2007*; *Tihelka et al., 2021*). Secondly, amber is known from the upper Campanian (~73 Ma) Pipestone Creek bonebed (Alberta, Canada) (*Tanke, 2004*; *Currie et al., 2008*). Although >99% of the 3000 individual fossils recovered from this site belong to *Pachyrhinosaurus*, other dinosaurs, fish, turtles, lizards, and crocodilians were also found (*Currie et al., 2008*; *Bell and Currie, 2016*; *Cockx et al., 2020*). Six bioinclusions recovered from ca. 50 cm$^3$ of typically <1 cm amber pieces were described (*Cockx et al., 2020*). Lastly, fossiliferous amber was found in Stratum 11 from the uppermost Maastrichtian (ca. 67–66 Ma) Bone Butte bonebed site (South Dakota, USA) (*DePalma, 2010*). This site, belonging to the intensively studied Hell Creek Formation, provided ~3000 mostly disarticulated fossils representing >50 species of dinosaurs and other vertebrates; the non-vertebrate material included molluscs, ichnofossils, and plant macroremains, and was mostly found together with the fossiliferous amber (*DePalma, 2010*; *DePalma et al., 2015*). The palaeodiversity recovered from the latter, in contrast, was rather scarce, with 22 bioinclusions found in 400 g of collected amber (*DePalma, 2010*; *DePalma et al., 2010*; *Nel et al., 2010*). Other Bone Butte strata yielded non-fossiliferous amber (*DePalma, 2010*). In addition, a hadrosaur jaw with an amber piece originally attached to it and containing an inclusion was reported from the uppermost Campanian Dinosaur Park Formation in Alberta (*McKellar et al., 2019*). Further Upper Cretaceous bonebed localities from western Canada yielded amber but lacking bioinclusions (*Cockx et al., 2021*).

The Ariño deposit represents one of the most important Lower Cretaceous dinosaur sites from Europe (*Alcalá et al., 2012*). This outcrop, located within the Santa María open-pit coal mine (Ariño municipality, Teruel Province, Spain), takes part in the Oliete Sub-basin of the Maestrazgo Basin (eastern Iberian Peninsula) (*Salas and Guimerà, 1996*). This extensional sub-basin was infilled with sediments deposited in palaeoenvironments ranging from marine to continental during the early Barremian to middle Albian (*Meléndez et al., 2000*). In this sub-basin, the siliciclastic Escucha Formation, early Albian in age (*Peyrot et al., 2007*; *Bover-Arnal et al., 2016*), was deposited overlying Aptian marine carbonates (*Cervera et al., 1976*). This formation represents coastal environments that included barrier-island systems with back-barrier marshes and flood-tidal deltas (*Rodríguez-López et al., 2009*). The AR-1 level of the Ariño locality, with ca. 600,000 m$^2$ of surveyed surface, consists of marls with a high concentration of organic matter occasionally forming coal, which underlie the lowest level of coal exploited in the Santa María mine (*Figure 1*, *Video 1*; *Alcalá et al., 2012*). The

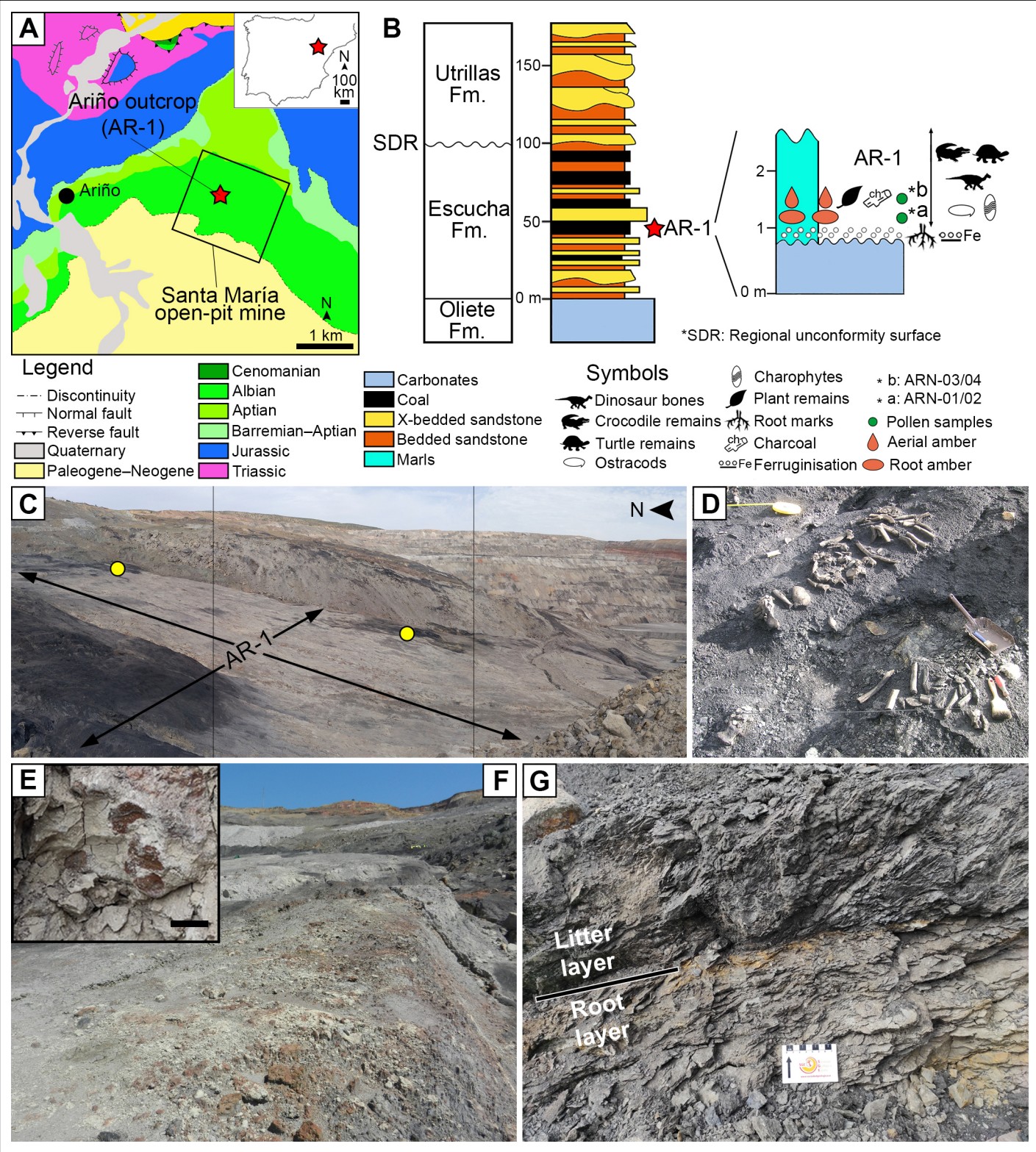

**Figure 1.** The Lower Cretaceous vertebrate bonebed and amber site of Ariño. (**A**) Geographical and geological location; modified from *Alcalá et al., 2012* (**B**) Stratigraphic location of the level AR-1; general stratigraphic log from the Oliete Sub-basin, modified from *Kirkland et al., 2013*, is shown at the left, together with the location of the level AR-1 (red star); a section of the latter, including the stratigraphic location of the amber deposit studied herein, is shown at the right. (**C**) Santa María open-pit coal mine with indication of the level AR-1 and the two excavated areas rich in aerial amber (yellow dots); the bottom of the open-pit coal is at the right. (**D**) One of the 160+ bone concentrations found in Ariño, AR-1/10, during vertebrate fieldwork in

*Figure 1 continued on next page*

*Figure 1 continued*

2010, showing the holotype of the nodosaurid *Europelta carbonensis*; metal dustpan ~30 cm long. (**E**) Root marks at the top of the carbonates below the level AR-1; scale bar, 1 cm. (**F**) Carbonates right below the level AR-1, displaying edaphic features at the top. (**G**) Detail photograph of the level AR-1 showing the lower root layer (with amber from resin exuded by roots) and the upper litter layer (with amber from resin exuded by trunk and branches); centimetric scale. See also *Video 1*.

AR-1 level has yielded a rich and diverse vertebrate fossil record representing more than 10,000 fossils namely found in more than 160 mono- or bitaxic concentrations of usually well-preserved, articulated or semi-articulated partial skeletons (*Alcalá et al., 2012*; *Alcalá et al., 2018*; *Buscalioni et al., 2013*; *Villanueva-Amadoz et al., 2015*). From these, new species of freshwater and terrestrial turtles, crocodilians, and ornithischian dinosaurs–that is, the ornithopod *Proa valdearinnoensis* and the nodosaurid *Europelta carbonensis*– have been described (*McDonald et al., 2012*; *Buscalioni et al., 2013*; *Kirkland et al., 2013*; *Pérez-García et al., 2015*; *Pérez-García et al., 2020*). Predatory dinosaurs were also present in the Ariño ecosystem, as evidenced by coprolites, ichnites, and isolated allosauroidean teeth (*Alcalá et al., 2012*; *Alcalá et al., 2018*; *Vajda et al., 2016*). Chondrichthyan and osteichthyan fish remains have also been occasionally found (*Alcalá et al., 2012*). Regarding the invertebrate record, three ostracod species (*Tibert et al., 2013*), as well as freshwater bivalves and gastropods, were reported (*Alcalá et al., 2012*; *Kirkland et al., 2013*). From the palaeobotanical standpoint, two charophyte species, fern remains, conifer twigs, taxonomically unassigned charcoalified wood remains, undetermined cuticles, and palynomorphs found in both the marls and coprolites (spores, gymnosperm, and angiosperm pollen grains) were previously known (*Tibert et al., 2013*; *Villanueva-Amadoz et al., 2015*; *Vajda et al., 2016*). Based on the former geological and palaeontological data, the Ariño palaeoenvironment was inferred as a freshwater swamp plain with perennial alkaline shallow lakes subjected to salinity fluctuations due to marine influence under a tropical–subtropical climate (*Alcalá et al., 2012*; *Tibert et al., 2013*; *Villanueva-Amadoz et al., 2015*). The level AR-1 was dated as early Albian (ca. 110 Ma) based on charophyte, palynological, and ostracod assemblages (*Tibert et al., 2013*; *Villanueva-Amadoz et al., 2015*; *Vajda et al., 2016*).

The presence of indeterminate amounts of amber in the AR-1 level from Ariño was first noted by *Alcalá et al., 2012*, with later works only adding that amber pieces were abundant and sometimes large (*Alcalá et al., 2018*). The only previously described bioinclusion from Ariño amber was a tuft of three remarkably well-preserved mammalian hair strands corresponding to the oldest hair reported in amber (*Álvarez-Parra et al., 2020a*).

In the Iberian Peninsula, amber is found in Triassic (Ladinian–Rhaetian) and Cretaceous (Albian–Maastrichtian) deposits; those having yielded abundant amber with bioinclusions are mostly late Albian in age, namely from the Basque-Cantabrian (e.g. Peñacerrada I and El Soplao) and Maestrazgo basins (e.g. San Just) (*Alonso et al., 2000*; *Delclòs et al., 2007*; *Peñalver et al., 2007*; *Najarro et al., 2009*; *Peñalver and Delclòs, 2010*).

Here, we characterise the amber deposit associated with the dinosaur bonebed AR-1 of Ariño from a multidisciplinary standpoint, describing its morphological, geochemical, palaeofaunistic, and taphonomic features, all of which allow us to recognise the palaeontological singularity of this site. Together with complementary palaeontological data (charophytes, palynomorphs, ostracods), our integrative results enable a complete reconstruction of the Ariño biota.

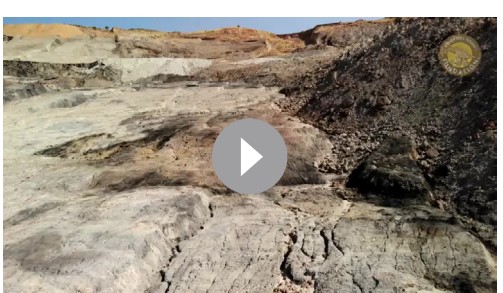

**Video 1.** Amber excavation in the lower Albian bonebed level AR-1 of Ariño during May 2019 and extraction of two strictly in situ (autochthonous) kidney-shaped amber pieces from the root layer. See also Figures 1 and 2.

https://elifesciences.org/articles/72477/figures#video1

## Results
### Amber characteristics

Two distinct amber-bearing layers, a lower one and an upper one, are present in the Ariño AR-1 level (*Figures 1 and 2*, *Figure 2—figure supplements 1 and 2*). The lower layer overlies a level of carbonates of oligotrophic lacustrine origin showing the development of palaeosols at its

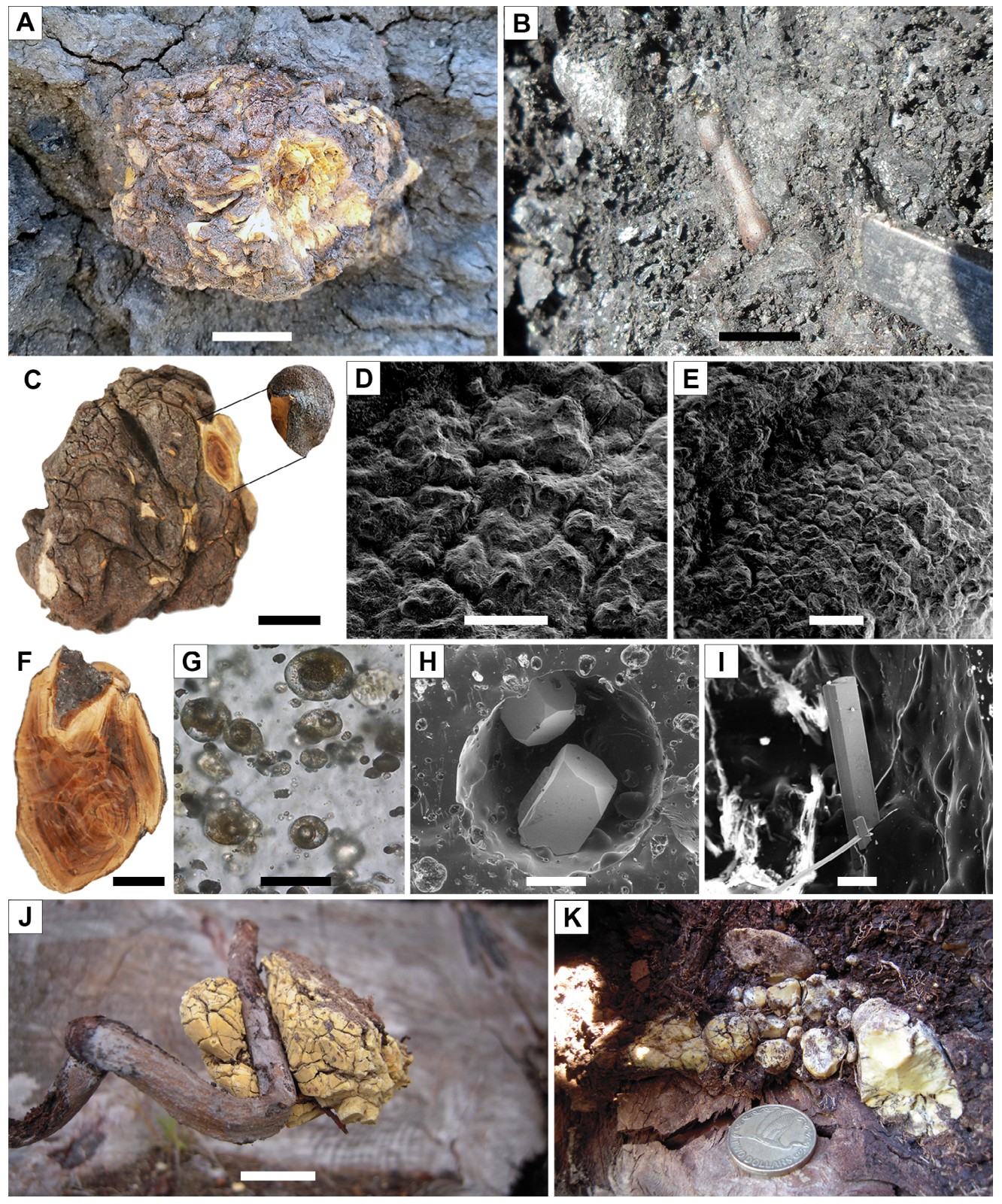

**Figure 2.** Diversity of amber pieces from the AR-1 level and Pleistocene copal pieces for comparison. (**A**) Kidney-shaped amber piece (root layer). (**B**) Aerial amber piece (litter layer), corresponding to a resin flow, after partially removing surrounding rock during fieldwork. (**C**) Kidney-shaped amber piece (AR-1-A-2019.93) from the root layer. (**D, E**) Two different areas of the external surface from a fragment detached from the piece in (**C**), showing the preserved delicate surface microprotrusions and no evidence of linear grooves. (**F**) Kidney-shaped amber piece (root layer) showing the internal banding

*Figure 2 continued on next page*

*Figure 2 continued*

pattern (AR-1-A-2019.132). (**G**) Triphasic (solid+ liquid + gas) bubble-like inclusions in a kidney-shaped amber piece (AR-1-A-2019.130). (**H**) Two pyrite cuboctahedrons in an alleged empty space left by a fluid inclusion (amber piece AR-1-A-2019.86). (**I**) Needle-shaped crystals from an iron sulphate (likely szomolnokite) growing inward from the walls in an alleged empty space left by a fluid inclusion (amber piece AR-1-A-2019.129). (**J**) Kidney-shaped piece of Pleistocene copal associated to an *Agathis australis* root from an overturned stump in Waipapakauri (North Island, New Zealand). (**K**) Pleistocene copal pieces associated to the root system of the same *A. australis* stump; coin 2.65 cm in diameter. Scale bars, 2 cm (**A–C, F, J**), 0.5 mm (**D**), 1 mm (**E**), 0.03 mm (**G**), 0.2 mm (**H**), and 0.1 mm (**I**). See also *Video 1*.

The online version of this article includes the following figure supplement(s) for figure 2:

**Figure supplement 1.** Strictly in situ (autochthonous) kidney-shaped amber pieces from the root layer of the lower Albian bonebed level AR-1 of Ariño.

**Figure supplement 2.** Litter layer of the lower Albian bonebed level AR-1 of Ariño.

**Figure supplement 3.** Amber pieces with taphonomic interest from the level AR-1 of Ariño.

top, including root marks (*Figure 1E and F*). This layer is characterised by abundant, irregular amber lumps (i.e., kidney-shaped) 10–40 cm in length with protrusions, an opaque crust, an inner banding pattern, and lacking bioinclusions (*Figure 2A, C and F*, *Figure 2—figure supplement 1*). Aerial amber and charcoalified plant remains are absent in this layer. The kidney-shaped amber pieces are distributed along the exposed area of the AR-1 level and, if not partially exposed due to weathering, are complete. The opaque crust from the amber pieces has an irregular morphology and its ultrastructure shows delicate microprotrusions and no evidence of linear grooves (*Figure 2C–E*). The banding patterns are formed by variable densities of abundant bubble-like inclusions of different sizes, which are monophasic (solid), biphasic (solid+ liquid), or triphasic (solid+ liquid + gas) (*Figure 2G*). Mineral crystals have been detected growing inwards within allegedly empty spaces left by larger bubble-like inclusions—these include pyrite cuboctahedrons and needle-shaped crystals from an iron sulphate mineral according to EDS analysis (likely szomolnokite, $Fe^{2+}SO_4 \cdot H_2O$) (*Figure 2H,I*, *Figure 2—figure supplement 3A*).

The upper layer from the Ariño AR-1 level is rich in amber pieces of flow-, droplet-, and stalactite-shaped morphologies, which often show external and/or internal desiccation surfaces (*Figure 2B*, *Figure 2—figure supplement 2*). Small, almost spherical amber pieces about 1–5 cm in diameter, with an opaque crust similar to the kidney-shaped amber pieces, are also present in this layer, yet rare; their surface is polished and more regular in patterning (*Figure 2—figure supplement 3B-D*). Amber pieces range from translucent to opaque, and from light yellow to dark reddish in colour. One peculiar piece showed subtle, multidirectional surface microscopic scratches and borings, the latter filled with an undetermined material, neither calcium carbonate nor gypsum (*Figure 2—figure supplement 3E*).

The FTIR spectra of two stalactite-shaped amber pieces from Ariño are dominated by a small C-H stretching band at 2925 cm$^{-1}$, an intense C-H band at 1457 cm$^{-1}$, and an intense carbonyl band at 1707 cm$^{-1}$ (*Figure 3A*, *Figure 3—source data 1*, *Figure 3—source data 2*, *Figure 3—source data 3*), all characteristic of amber (*Grimalt et al., 1988*). Hydroxyl bands near 3500 cm$^{-1}$ are present. The Ariño amber spectra are very similar to those from San Just amber, their main difference being the presence of a small band near 1200 cm$^{-1}$ in the latter. On the other hand, the composition of the organic solvent-extractable materials obtained by GC-MS, comprising the 32.5 % of the Ariño amber, is dominated by labdane resin acids and its diagenetic derivatives, with amberene (**I**; 1,6-dimethyl-5-isopentyltetralin) being the major component in the bulk extract (*Figure 3B*, *Figure 3—figure supplement 1*, *Figure 3—source data 4*). The labdan-18-oic acids are dominant in the polar fraction of the organic extract from the amber. The identification of the clerodane-family diterpene **VI** is noteworthy. The analyses show no evidence of significant terpenes of the pimarane/abietane family and discount the presence of ferruginol. The Ariño amber does not show a significant content of either 15-homoamberene (**III**) or 1-methylamberene (**X**) (*Figure 3B*, *Figure 3—figure supplement 1*; *Kawamura et al., 2018*). This could point to a lack of the corresponding labdanoid alcohols or non-oxidised C18/C19 labdanoids in the precursor resin, as the diagenesis of these molecules could lead to 1-methylamberene. The decarboxylation of the labdan-18-oic acids prevailing in the Ariño amber polar fraction could be the first step in the diagenesis to amberene and its related compounds, especially isomers of the labdanoid **VI**, found as a rich distribution of peaks with M$^+$=246.

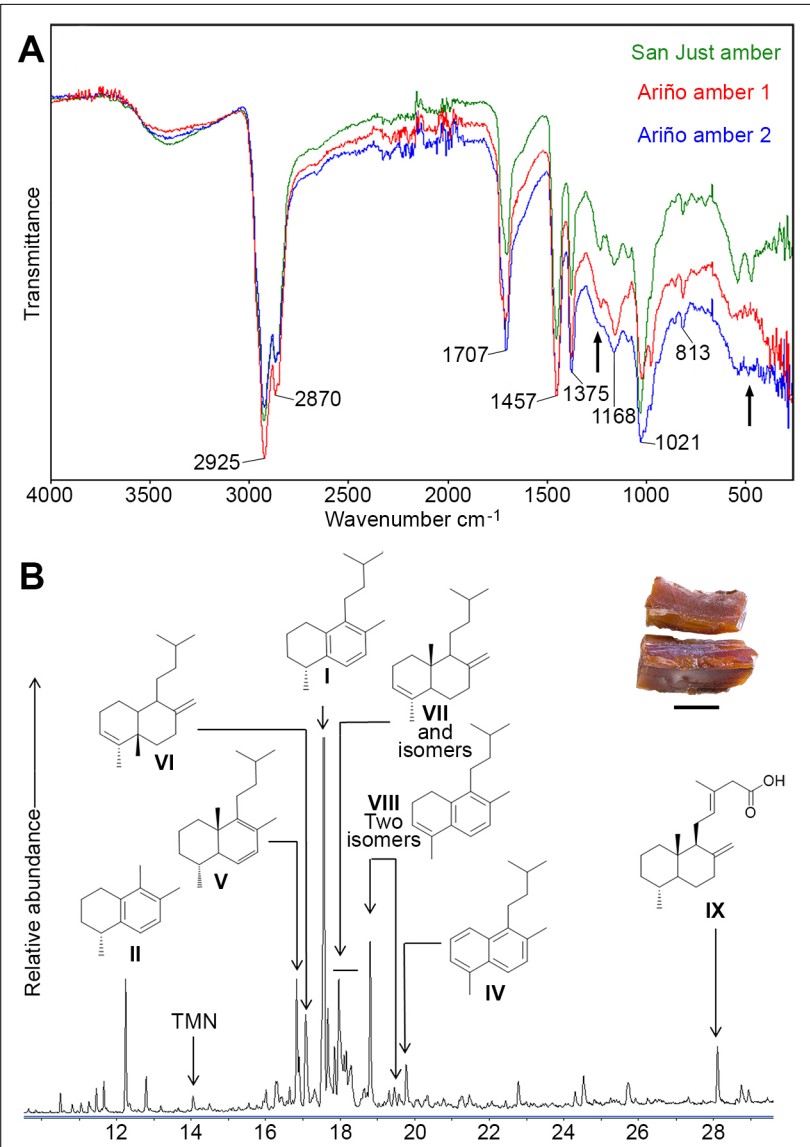

**Figure 3.** Physicochemical characterisation of the Lower Cretaceous amber from Ariño. (**A**) Infrared (FTIR) spectra obtained from two aerial amber pieces (litter layer); a spectrum from San Just amber (upper Albian) is provided for comparison; arrows indicate the main differences between Ariño and San Just ambers, at around 1200 and 500 cm⁻¹; resolution = 4 cm⁻¹. (**B**) Gas chromatography-mass spectrometry (GC-MS) trace for the underivatised total solvent extract of aerial amber, showing the structures of the main identified terpenoids, referred herein using Roman numerals (full formulation provided in *Figure 3—figure supplement 1B*); TMN = trimethylnaphthalenes; the analysed aerial amber is shown at the top right (scale bar 0.5 mm).

The online version of this article includes the following figure supplement(s) for figure 3:

**Source data 1.** FTIR data of the Ariño amber 1.

**Source data 2.** FTIR data of the Ariño amber 2.

**Source data 3.** FTIR data of the San Just amber.

**Source data 4.** GC-MS data of the Ariño amber.

**Figure supplement 1.** Physicochemical characterisation of the Lower Cretaceous amber from Ariño.

## Bioinclusions

A total of 166 bioinclusions were obtained out of 918 g of aerial amber (*Figure 4*, *Figure 4—figure supplement 1*, *Figure 4—figure supplement 2*); about one third of them are well to exceptionally well preserved. Plant inclusions are present, such as numerous fern or conifer trichomes (not considered in the inclusion count) and other undetermined remains (*Figure 4—figure supplement 1A-E*). The diverse assemblage is chiefly composed of arthropods or remains of their activity, such as spiderweb threads (*Figure 4—figure supplement 1F,G*) and coprolites, but also a few vertebrate integumentary remains. Arthropods are represented by arachnids and 11 insect orders. Arachnid inclusions belong to mites (Acari) and spiders (Araneae). Mites include a rare trombidiform of the family Rhagidiidae, an oribatid, and an undetermined six-legged larva (*Figure 4A*). One spider offers taphonomic insights (*Figure 4—figure supplement 1H*). Five amber pieces with arthropods as syninclusions have spiderwebs preserved; although all are isolated strands, one tangled sample might correspond to a partial web (*Figure 4—figure supplement 1F*). In the latter, glue droplets on several strands suggest it belonged to an orb web (*Figure 4—figure supplement 1G*). The insect orders found in the Ariño amber are jumping bristletails (Archaeognatha), crickets (Orthoptera), cockroaches (Blattodea), barklice (Psocodea), thrips (Thysanoptera), whiteflies and aphids (Hemiptera), lacewings (Neuroptera), beetles (Coleoptera), moths (Lepidoptera), gnats, midges, and other flies (Diptera), and wasps (Hymenoptera). Archaeognaths are represented by the inclusion of a cercus and a medial caudal filament. Two orthopterans are poorly preserved, but one could belong to †Elcanidae. A blattodean nymph and an adult have been found, as well as several remains such as probably blattodean isolated antennae. Among the seven psocodeans discovered, new taxa probably within the †Archaeatropidae and Manicapsocidae have been recognised. Thysanopterans are the third most abundant insect order in the Ariño amber, with 11 specimens (*Figure 4B and C*); three amber pieces contain more than one thrips as syninclusions. One isolated thrips shows a thin milky coating (*Figure 4—figure supplement 1I*), also found in other inclusions, and an infrequent nymph is unusually well preserved (*Figure 4B*). Hemipterans comprise four representatives of Sternorrhyncha and two incomplete undetermined specimens. Three of the former have been identified as Aleyrodidae, probably belonging to the Aleurodicinae (*Figure 4D*), and are preserved in the same amber piece as syninclusions. In addition, an Aphidoidea specimen (*Figure 4—figure supplement 2A*) has been found in an amber piece with spiderweb strands. The neuropteran record consists of two wing impressions on amber surfaces probably belonging to Berothidae (*Figure 4—figure supplement 2B*) and a complete specimen, which could correspond to a †Paradoxosisyrinae (Sisyridae) (*Figure 4—figure supplement 2C*). Five coleopteran specimens have been discovered, two tentatively identified as belonging to Ptinidae (*Figure 4—figure supplement 2D*) and Cantharidae (*Figure 4—figure supplement 2E*). A ditrysian lepidopteran larva, yet incomplete anteriorly, is remarkably well preserved (*Figure 4E*). Dipterans (*Figure 4F*) are represented by 19 specimens of the families †Archizelmiridae, Cecidomyiidae, Ceratopogonidae (including at least one female), Chironomidae, Mycetophilidae, Rhagionidae, Scatopsidae, and probably Psychodidae. The first group is represented by a well-preserved male within the genus *Burmazelmira* (*Figure 4F*). Lastly, hymenopterans are the most abundant insects in Ariño amber, accounting for 34 specimens belonging to the Platygastroidea, Mymarommatoidea, †Serphitidae, and †Stigmaphronidae (*Figure 4G–I*). Furthermore, a new vertebrate inclusion is represented by a basal feather barb portion with a pennaceous structure (*Figure 4J*).

## Palaeobotanical and ostracod assemblages

Charophytes sampled from the level AR-1 comprise four species belonging to the families †Clavatoraceae and Characeae. The assemblage is dominated by †Clavatoraceae, particularly by well-preserved fructifications of *Atopochara trivolvis* var. *trivolvis* (*Figure 5A-D*, *Figure 5—figure supplement 1A,B*) and *Clavator harrisii* var. *harrisii* (*Figure 5E-J*, *Figure 5—figure supplement 1C-E*) (n > 100 for each species). The former is represented by large utricles showing a characteristic triradiate symmetry and displaying flame-shaped cells at positions a1, a3, b1, and c1 (*Grambast, 1968*); such a configuration is variable in other populations of the same species. Five morphotypes of *C. harrisii* var. *harrisii* have been distinguished based on the cell disposition at the abaxial side of the utricle, showing the phylloid imprint flanked at the base by two small cells and bearing above a complex of 5–11 cells (*Figure 5—figure supplement 1F-J*); although this variety had been previously identified in Ariño, none of the morphotypes described herein for the first time in the variety were evident (*Tibert et al., 2013*).

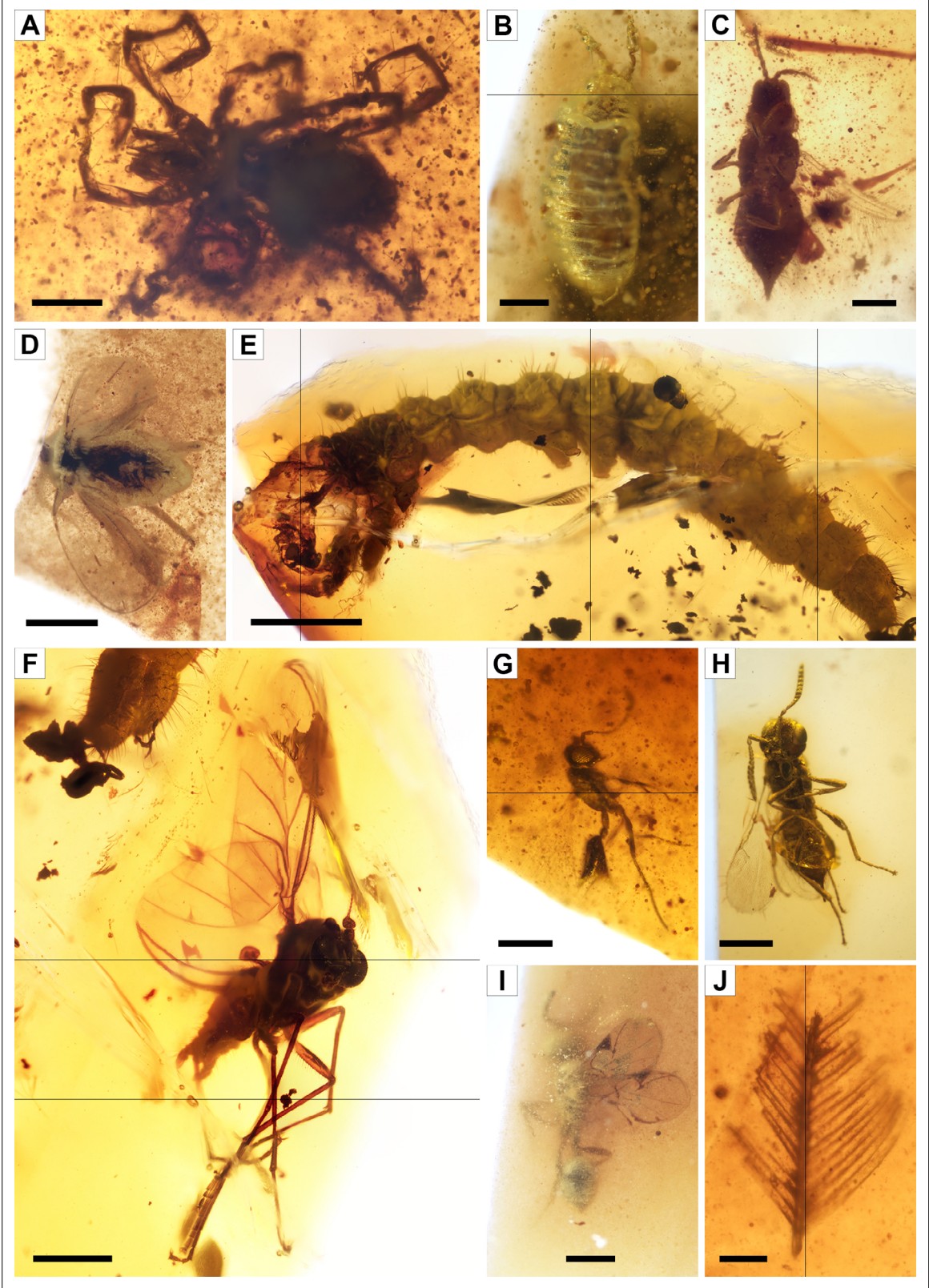

**Figure 4.** Faunal bioinclusions from the Lower Cretaceous bonebed amber of Ariño. (**A**) A rhagidiid mite, the oldest known (Acari: Rhagidiidae; AR-1-A-2019.71). (**B**) An immature thrips (Thysanoptera; AR-1-A-2019.114.2). (**C**) An adult thrips (Thysanoptera; AR-1-A-2019.40). (**D**) A whitefly (Hemiptera: Aleyrodidae; AR-1-A-2019.100.1). (**E**) A ditrysian lepidopteran larva (AR-1-A-2019.95.1). (**F**) A *Burmazelmira* sp. fly (Diptera: †Archizelmiridae: AR-1-A-2019.95.2). (**G**) A false fairy wasp, the oldest known (Hymenoptera: Mymarommatoidea; AR-1-A-2019.61). (**H**) A superbly preserved platygastroid wasp

*Figure 4 continued on next page*

*Figure 4 continued*

(Hymenoptera: Platygastroidea; AR-1-A-2019.95.3). (**I**) A serphitid wasp, the oldest known (Hymenoptera: †Serphitidae; AR-1-A-2019.94.8). (**J**) A feather barb fragment with pennaceous structure (Theropoda; AR-1-A-2019.53). Scale bars, 0.2 mm (**A–C, G**), 0.5 mm (**D, F, H, I**), 1 mm (**E**), and 0.1 mm (**J**).

The online version of this article includes the following figure supplement(s) for figure 4:

**Figure supplement 1.** Diverse bioinclusions in amber from the level AR-1 of Ariño.

**Figure supplement 2.** Insect bioinclusions in amber from the level AR-1 of Ariño.

Moreover, several (n = 10) portions of clavatoracean thalli belonging to *Clavatoraxis* sp. have been recovered (*Figure 5K*). Lastly, rare occurrences (n = 3) of small characean gyrogonites with affinities to *Mesochara harrisii* are also present (*Figure 5L*); their determination remains somewhat uncertain due to the lack of a basal plate (*Martín-Closas et al., 2018*).

Charcoalified plant remains (fusinite/inertinite) are abundant in the upper amber layer (*Figure 2—figure supplement 2A*). These correspond to secondary xylem with strongly araucariacean (1) 2 (3) seriate intertracheary radial pitting and araucarioid cross-fields. Although they are similar to the araucariacean *Agathoxylon gardoniense*, we prefer to identify these samples as *Agathoxylon* sp. due to preservation. Other charcoalified wood remains likely belonging to other taxonomic groups have also been found. Furthermore, a rare sample of amber-filled plant tissue shows cells elliptic to rounded in cross-section and elongate in longitudinal section, blunt tips, 20–50 µm in diameter, and with thin walls (somewhat collenchymatous). Cells are arranged radially, but without evidence of growth rings. These characteristics suggest that this fossilised tissue might represent suber (cork) (*Figure 4—figure supplement 1B-E*).

The four studied palynological samples (ARN-01–ARN-04) have provided highly diverse, well-preserved assemblages that include a total of 72 different palynomorph taxa, that is, two from freshwater algae, 38 from spores of ferns and allied groups, 21 from gymnosperm pollen grains, and 11 from angiosperm pollen grains (*Figure 5M-T*, *Figure 5—figure supplement 1K-P*, *Supplementary file 1*). Aquatic palynomorphs, consisting of zygnematacean freshwater algae, are a small proportion of the samples except for ARN-04, characterised by the abundance of *Chomotriletes minor* (3.87 % of the total palynomorph sum) (*Figure 5M*). Spores numerically dominate the assemblages except for ARN-01. Overall, fern spores such as *Appendicisporites* spp. (*Figure 5N*), *Cicatricosisporites* spp., *Cyathidites australis*, *Cyathidites minor* (*Figure 5O*), and *Gleicheniidites senonicus*, predominate (13.48–38.42%) over those of bryophytes and lycophytes (1.44–2.38%). Gymnosperms are namely represented by *Inaperturopollenites dubius* (12.79–20.36%) (*Figure 5P*), related to taxodioid conifers (*Stuchlik et al., 2002*), and the genus *Classopollis* (9.39–15.95%) (*Figure 5Q*), produced by †Cheirolepidiaceae conifers (*Taylor and Alvin, 1984*). Araucariacean and bisaccate pollen show low amounts except for the araucariacean *Araucariacites* spp. (*Figure 5R*), which is particularly abundant in ARN-01 (12.79%). The abundance of *Eucommiidites* spp., assigned to †Erdtmanithecaceae gymnosperms, is also relevant (2.90%–8.00%) (*Figure 5S*). '*Liliacidites*' *minutus* (*Figure 5T*) was the most abundant angiosperm pollen in the assemblages (up to 12 % in ARN-01).

The ostracod fauna recovered from the level AR-1 is comprised of four species belonging to the families Limnocytheridae, †Cyprideidae, and Cyprididae (*Figure 5U–EE*, *Figure 5—figure supplement 1Q–Y*). Specimens show mostly closed carapaces and are generally well preserved. The Limnocytheridae are represented by *Theriosynoecum* cf. *fittoni* (*Figure 5U and V*) (n = 20) and *Rosacythere denticulata* (n > 70) (*Figure 5W–BB*). Although the latter species was previously identified in Ariño (*Tibert et al., 2013*), three variants have now been detected: one with a faint pitting and extremely small rosette ornamentation (*Figure 5W*), one with a well-developed rosette (*Figure 5X, Y and AA*, *Figure 5—figure supplement 1Q–U*), and one with strongly developed rosette and spine-like nodes locally generated at the postero-dorsal and postero-ventral parts of the carapace (*Figure 5Z and BB*, *Figure 5—figure supplement 1V–Y*). †Cyprideidae and Cyprididae are found for the first time in Ariño, represented by the species *Cypridea* cf. *clavata* (n = 35) (*Figure 5CC and DD*) and *Mantelliana* sp. (n = 12) (*Figure 5EE*), respectively.

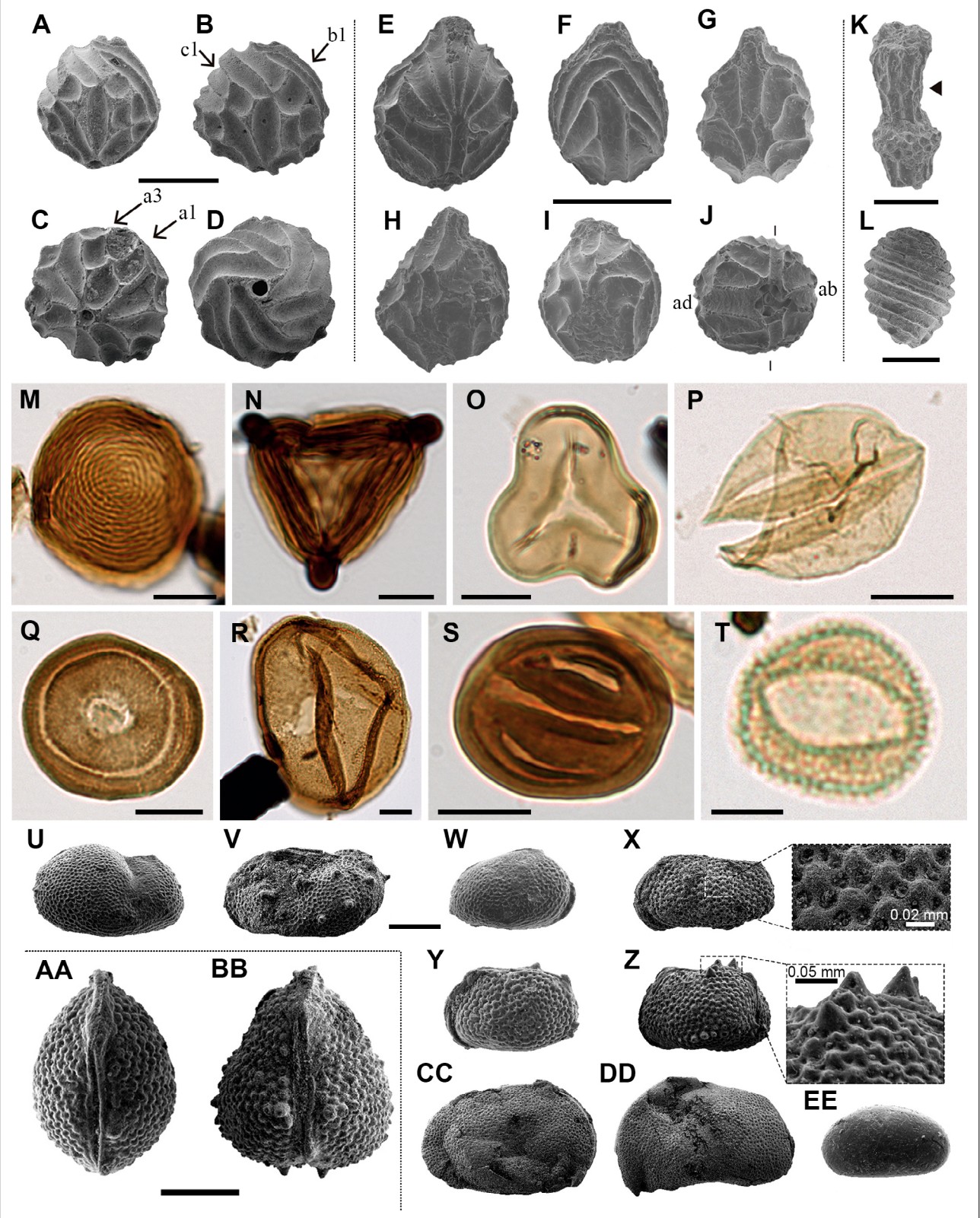

**Figure 5.** Charophyte (A–L), palynomorph (M–T), and ostracod (U–EE) records sampled from level AR-1 of Ariño. (**A–D**) *Atopochara trivolvis* var. *trivolvis* (†Clavatoraceae): (**A, B**) Lateral views (AR-1-CH-004 and AR-1-CH-005, respectively); (**C**) Basal view (AR-1-CH-007); (**D**) Apical view (AR-1-CH-008); cell lettering after *Grambast, 1968*. (**E–J**) *Clavator harrisii* var. *harrisii* (†Clavatoraceae): (**E**) Lateral view (AR-1-CH-009); (**F**) Adaxial view (AR-1-CH-011); (**G**) Abaxial view morphotype II (AR-1-CH-013); (**H**) Abaxial view morphotype III (AR-1-CH-014); (**I**) Abaxial view morphotype IV (AR-1-CH-015); (**J**) Basal

*Figure 5 continued on next page*

*Figure 5 continued*

view (AR-1-CH-017) with indication of adaxial (ad) and abaxial (ab) sides. (**K**) *Clavatoraxis* sp. (†Clavatoraceae) (AR-1-CH-019); the arrowhead indicates the zig-zag line at the central part of the internode. (**L**) aff. *Mesochara harrisii* (Characeae) in lateral view (AR-1-CH-001). (**M**) *Chomotrileres minor* (ARN-03). (**N**) *Appendicisporites tricornitatus* (ARN-01). (**O**) *Cyathidites minor* (ARN-02). (**P**) *Inaperturopollenites dubius* (ARN-04). (**Q**) *Classopollis* sp. (ARN-02). (**R**) *Araucariacites hungaricus* (ARN-01). (**S**) *Eucommiidites minor* (ARN-01). (**T**) "*Liliacidites*" *minutus* (ARN-01). (**U, V**) *Theriosynoecum* cf. *fittoni* (Limnocytheridae): (**U**) Right lateral view, female carapace (AR-1-OS-001); (**V**) Left lateral view, male carapace (AR-1-OS-002). (**W–BB**) *Rosacythere denticulata* (Limnocytheridae): (**W**) Female? carapace, right lateral view, variant with extremely small rosette ornamentation (simply reticulated form) (AR-1-OS-006); (**X**) Male carapace of the variant with well-developed rosette ornamentation, left lateral view (AR-1-OS-011), and detail of the ornamentation; (**Y**) Female carapace of the variant with well-developed rosette ornamentation, left lateral view (AR-1-OS-007); (**Z**) Female carapace of the variant with strongly developed rosette ornamentation and spine-like nodes, left lateral view (AR-1-OS-015), and detail of the spine-like node ornamentation; (**AA**) Female carapace of the variant with well-developed rosette ornamentation, dorsal view (AR-1-OS-012); (**BB**) Female carapace of the variant with strongly developed rosette ornamentation and spine-like nodes, dorsal view (AR-1-OS-018), showing intraspecific variability. (**CC**), (**DD**) *Cypridea* cf. *clavata* (†Cyprideidae): (**CC**) Specimen in right lateral view (AR-1-OS-004); (**DD**) Specimen in left lateral view (AR-1-OS-005). (**EE**) *Mantelliana* sp. (Cyprididae) (AR-1-OS-003), right lateral view. Scale bars, 0.5 mm (**A–J**), 0.25 mm (**K**), 0.2 mm (**L, U–EE**), 0.01 mm (**M–S**), and 0.005 mm (**T**). See also *Supplementary file 1*.

The online version of this article includes the following figure supplement(s) for figure 5:

**Figure supplement 1.** Additional charophyte (A–J), palynomorph (K–P), and ostracod (Q–Y) records sampled from level AR-1 of Ariño.

## Discussion
### Taphonomy

The kidney-shaped, bioinclusion-lacking amber pieces from AR-1's lower amber-bearing layer were produced by roots. Subterranean accumulations from both Recent/subfossil resin in modern forests and amber in geological deposits have been partly attributed to roots (*Langenheim, 1967*; *Langenheim, 2003*; *Henwood, 1993*; *Martínez-Delclòs et al., 2004*; *Seyfullah et al., 2018*). Although the resiniferous capacity of roots is well known (*Langenheim, 2003*), observations of resin attached and/or associated with roots from both angiosperms and gymnosperms have been occasional (*Langenheim, 1967*; *Seyfullah et al., 2018*). Our field observations of late Pleistocene copal pieces produced and still attached to roots, covered by original soil, in an *Agathis australis* overturned stump, which was formerly referred to but not figured (*Najarro et al., 2009*; *Speranza et al., 2015*), show similar morphologies to the Ariño kidney-shaped amber pieces (*Figure 2J and K*). The Ariño's lower amber layer is interpreted as a root layer where the abundant and complete amber pieces are strictly in situ, that is, they are located exactly where the roots of the resiniferous trees exuded this resin in the subsoil (*Figure 6A*). This level immediately overlies carbonates that display edaphic features at the top (*Figure 1E and F*). It has high lateral continuity and lacks aerial amber or charcoalified plant remains (*Figures 1 and 2*, *Figure 2—figure supplement 1*). Also, the fragile surface protrusions and microprotrusions of the kidney-shaped amber pieces from this layer would not have preserved even if minimal biostratinomic transport or other processes entailing abrasion had occurred (*Figure 2C–E*). This is the first time strictly in situ amber is reported; the scarcely fossiliferous, autochthonous-parautochthonous Triassic amber droplets from the Dolomites are preserved in a palaeosol (*Schmidt al., 2012*; *Seyfullah et al., 2018*), but they are strictly ex situ, as the resin at least fell by gravity from their aboveground exudation location to the forest floor (*Figure 6A*). Although other amber-bearing outcrops from the Iberian Peninsula have commonly yielded kidney-shaped amber pieces (*Alonso et al., 2000*; *Peñalver et al., 2007*; *Najarro et al., 2009*), these appear fragmented and in pockets together with aerial amber and generally have smoother surfaces and more regular morphologies than those from Ariño. The kidney-shaped amber pieces have further noteworthy characteristics. Firstly, the amber pieces show marked internal bands composed of variable densities of mono-, bi-, or triphasic bubble-like inclusions (*Figure 2F and G*). Although these microscopic inclusions likely correspond to fossilised sap-resin emulsions (*Lozano et al., 2020*), at least partially, they show more complex and previously undocumented morphologies and arrangements. These microinclusions have the potential to provide key data on taphonomy and the conditions under which resin production occurred. Moreover, pyrite cuboctahedrons (*Figure 2H*) are usually found as mineralisations in the alleged empty spaces left by fluid bubble-like inclusions within amber (*Alonso et al., 2000*); they have been related to early diagenesis in reducing environments produced by anaerobic bacteria (*Allison, 1990*). In contrast, the iron sulphate minerals growing in these spaces (*Figure 2I*, *Figure 2—figure supplement 3A*) have not been previously reported; they could have formed during late diagenesis under oxidising

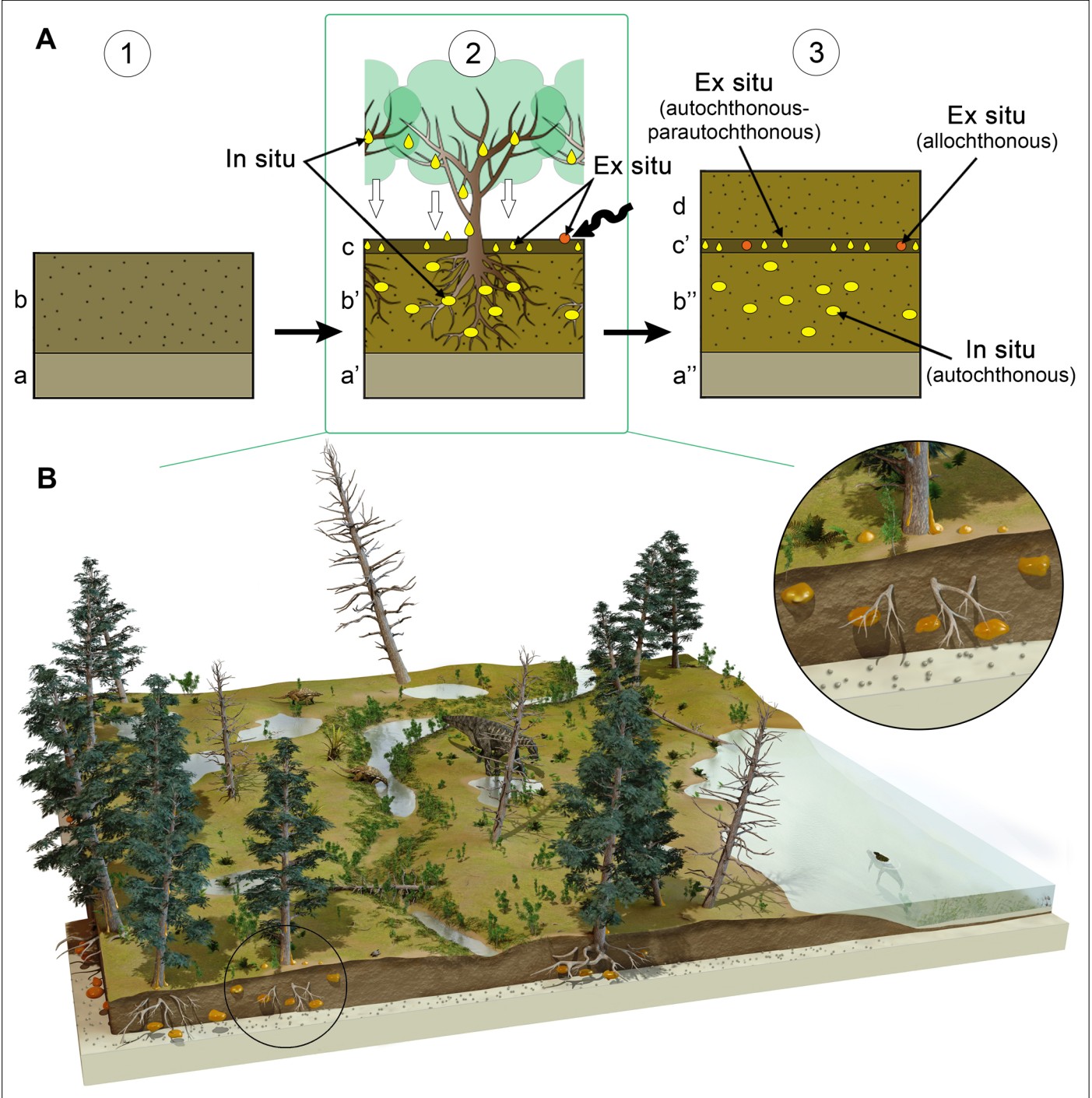

**Figure 6.** Formation of the amber deposit of Ariño. (**A**) Idealised diagrams depicting (1) the original depositional environment (a, carbonates; b, soil prior to tree installation); (2) resiniferous forest installation and pedogenesis; concentration of in situ kidney-shaped resin pieces produced by the roots in a root horizon (b'); accumulation of aerial resin pieces fallen from the branches and trunk and a few resin pieces dragged after transport (wavy arrow) in a litter horizon (c); and (3) fossil diagenesis of the resin pieces, resulting in a layer containing strictly in situ autochthonous kidney-shaped amber pieces produced by roots (b''), and a layer mostly composed of strictly ex situ autochthonous-parautochthonous aerial amber pieces and a few potentially allochthonous amber pieces (c'); level AR-1 corresponds to a single cycle of forest floor installation-destruction. (**B**) Artistic reconstruction of the coastal freshwater swamp ecosystem of Ariño, with emphasis on the depositional environment of the resin. The resiniferous trees are araucariaceans (extant model used: *Agathis australis*), tentatively identified as the resin source of Ariño; other depicted terrestrial plants are undetermined vegetation included for artistic purpose. Charophytes and a crocodile (*Hulkepholis plotos*) inhabit the shallow water body on the right; two nodosaurids (*Europelta carbonensis*), an iguanodontian (*Proa valdearinnoensis*), and a turtle (*Aragochersis lignitesta*) are shown on land; these vertebrate species were erected based on the Ariño bonebed material. Artist of the illustration in (**B**): José Antonio Peñas.

conditions after the input of oxygenated water into the amber (*Allison, 1990*). Secondly, the kidney-shaped amber pieces from Ariño lack the coating of resinicolous fungal mycelia otherwise common in Cretaceous ambers (*Speranza et al., 2015*). Both the mono- to multiphasic inclusions and the lack of fungal coating might be related to the likely partial flooding of the Ariño forest soil, typical of swampy environments. Although resin can solidify on or within the forest soil in tropical or subtropical climate environments (*Henwood, 1993*), plant macrofossils are usually highly altered and poorly preserved in soils and coal deposits (*Delclòs et al., 2020*). The absence of roots associated with the kidney-shaped amber pieces in the root layer could be explained by their differential fossilisation in the partly flooded soil. The inside-out growth pattern of the resin produced by roots, based on our field observations, is incompatible with the inclusion of surrounding soil particles in the outer surface of the resin, as the opaque crust readily hardens and pushes back the sediment around it.

The upper amber-bearing layer from Ariño's AR-1 level is rich in aerial amber pieces (*Figure 2B*, *Figure 2—figure supplement 2*). This amber type results from fluid resin falling on the ground from the trunk or branches (*Martínez-Delclòs et al., 2004*). The aerial amber pieces commonly have delicate morphologies and preserve external desiccation surfaces, both elements indicating very limited transport. Overall, the upper layer is interpreted as a litter layer namely resulting from the autochthonous-parautochthonous accumulation of strictly ex situ aerial amber pieces, but also occasionally containing amber pieces showing surface polishing or scratching and thus likely being more allochthonous in nature, that is, transported and deposited far away from their production environment (*Figure 6A*). Moreover, the absence of strictly in situ kidney-shaped amber pieces in the litter layer suggests that AR-1 corresponds to a single cycle of forest floor installation-destruction. Charcoalified wood remains are abundant in this layer; these were previously found in Ariño and were related to wildfires (*Villanueva-Amadoz et al., 2015*; *Vajda et al., 2016*), which have been deemed as promoters of resin production and accumulation (*Najarro et al., 2010*; *Seyfullah et al., 2018*). On the other hand, the aerial amber is highly fossiliferous, with 145 bioinclusions/kg (excluding coprolites, spiderwebs, and undetermined bioinclusions). Although determining amber bioinclusion richness is prone to multiple biases (e.g., bioinclusion occurrence data should be ideally limited to aerial amber, as kidney-shaped amber fragments were almost certainly devoid of bioinclusions), this value is among the highest reported worldwide. Richness data from other Cretaceous Albian to Cenomanian amber localities range from about 10–80 inclusions/kg (*Grimaldi et al., 2002*; *Néraudeau et al., 2002*; *Girard et al., 2013*; *Peñalver et al., 2018*; *Zheng et al., 2018*), although values surpassing the 500 insects/kg have been exceptionally reported (*Rasnitsyn and Quicke, 2002*). A few bioinclusions are covered by a white foam consisting of microscopic bubbles produced by decomposition fluids during early diagenesis (*Figure 4—figure supplement 1I*), similar to that commonly observed in the Eocene Baltic amber (*Martínez-Delclòs et al., 2004*), but otherwise rare among Cretaceous ambers.

The charophyte, palynological, and ostracod data provided herein are also indicative of a very limited transport of these remains prior to burial. The utricles of the two clavatoracean charophytes found are well preserved and abundant, suggesting that these remains are autochthonous or parautochthonous. The occurrence of clavatoracean portions of thalli associated with the fructifications supports this inference. The studied palynological samples show conspicuous abundances of pollen related to araucariacean trees and angiosperms in ARN-01 (root layer), which could indicate parautochthony based on their limited pollen production and dispersal potential (*Taylor and Hu, 2010*). Samples ARN-03 and ARN-04 (litter layer) contain low araucariacean and angiosperm pollen and high amounts of allochthonous wind-transported miospores such as *Cyathidites* spp., *Inaperturopollenites dubius*, and *Classopollis* spp., which suggest chiefly allochthonous assemblages. In contrast, ARN-02 (at the top of the root layer) shows a parautochthonous-allochthonous transitional assemblage based on an increase of fern spores and erdtmanithecalean pollen, as well as lower values of araucariacean pollen than ARN-03 and ARN-04. The previous palaeobotanical accounts from Ariño concluded parautochthony based on the good palynomorph preservation, with some samples even showing their original tetrad configuration, although they recognised that some charcoalified wood remains could be allochthonous (*Villanueva-Amadoz et al., 2015*; *Vajda et al., 2016*). Lastly, the studied ostracods constitute a relatively rich assemblage characterised by abundant specimens with closed carapaces (*Figure 5AA and BB*), which together with the low percentage of broken individuals points towards autochthonous remains (*Trabelsi et al., 2021*).

The previous taphonomic accounts on the Ariño vertebrates indicated the absence or lowest grade of biostratinomic transport. The abundant vertebrate fossils are namely found in monotaxic (occasionally bitaxic) concentrations of well-preserved, articulated, or semi-articulated remains (*Alcalá et al., 2012*; *Alcalá et al., 2018*; *Buscalioni et al., 2013*; *Villanueva-Amadoz et al., 2015*). Coprolites, likely dinosaurian, show a palynomorph composition similar to that of the rock (*Vajda et al., 2016*). By integrating all the taphonomic data from the diverse palaeobiological elements from Ariño, we can conclude that the great majority of the assemblage, except for some pollen and charcoalified plant material, as well as a small percentage of the amber, had an autochthonous or parautochthonous origin, and therefore roughly inhabited or was produced in the same area where it fossilised. This circumstance, although critical for inferring reliable data on the palaeoecosystem (*Martínez-Delclòs et al., 2004*), remains infrequent among palaeontological deposits, particularly those jointly preserving dinosaur remains and fossiliferous amber. Indeed, the three previously reported localities where fossiliferous amber was found associated with dinosaur bonebeds, all from the Late Cretaceous, show clear signs of being either allochthonous or clearly mixed assemblages in which at least a substantial part of the vertebrate remains suffered significant transport prior to burial (*Néraudeau et al., 2003*; *Currie et al., 2008*; *DePalma, 2010*): (1) Fouras/Bois Vert (="Plage de la Vierge") was interpreted as resulting from a catastrophic event such as a storm in a coastal estuarine environment, with the fragmentary bones showing evidence of considerable pre-burial transport; however, amber was assumed to be not heavily transported due to the lack of rounding (*Néraudeau et al., 2003*); (2) the Pipestone Creek monodominant vertebrate assemblage corresponds to disarticulated bones formed by a fluvial allochthonous accumulation in a vegetated floodplain, and interpreted as a mass mortality event; no taphonomic assessment for the amber was provided (*Tanke, 2004*; *Currie et al., 2008*; *Cockx et al., 2020*); and (3) Bone Butte's Stratum 11 was reconstructed as a mixed (~70/30) autochthonous/allochthonous vertebrate assemblage deposited in a river oxbow lake; amber showed no signs of significant transport (*DePalma, 2010*). From the diagenetic standpoint, a high maturity of the Ariño amber samples is inferred based on the absence of exocyclic methylenic bands at 880 cm$^{-1}$, 1640 cm$^{-1}$, and 3070 cm$^{-1}$ in the FTIR spectra, in accordance with their Cretaceous age (*Grimalt et al., 1988*). Furthermore, there is no significant difference in the distribution of trimethylnaphthalene isomers between the Ariño amber and the other ambarene-rich Cretaceous Iberian ambers in the GC-MS analyses (*Menor-Salván et al., 2016*), suggesting a similar thermal and diagenetic history (*Strachan et al., 1988*).

## Age of the level AR-1

Our charophyte, palynological, and ostracod data support the dating of the level AR-1 as early Albian in age (around 110 Ma), as previously proposed for Ariño and, more generally, the whole Escucha Formation (*Alcalá et al., 2012*; *Tibert et al., 2013*; *Villanueva-Amadoz et al., 2015*; *Bover-Arnal et al., 2016*; *Vajda et al., 2016*). The whole timespan of the co-occurrence of the charophytes *Atopochara trivolvis* var. *trivolvis* and *Clavator harrisii* var. *harrisii* is late Barremian–early Albian. However, in the late Barremian–early Aptian timespan, these species are associated to *A. trivolvis* var. *triquetra* (*Pérez-Cano et al., 2020*). Based on the occurrence of homogeneous populations of *A. trivolvis* var. *trivolvis*, the studied assemblage is assigned to the upper Aptian–lower Albian European Clavator grovesii var. corrugatus ( = Clavator grovesii var. lusitanicus) biozone of *Riveline et al., 1996*. This view is complementary to that based on the previously found co-occurrence in Ariño of *Clavator harrisii* var. *harrisii* and *Clavator harrisii* var. *zavialensis* indicating an early Albian age (*Tibert et al., 2013*). The oldest occurrence of *A. trivolvis* var. *trivolvis* has recently been reported from the upper Barremian (*Pérez-Cano et al., 2020*), but this variety is more characteristic of upper Aptian–Albian deposits (*Martín-Closas, 2000*). Regarding the palynomorphs, and in accordance with the age inferred by *Peyrot et al., 2007*, the occurrence of *Retimonocolpites dividuus* (*Figure 5—figure supplement 1N*) in ARN-03 indicates an age not older than late Aptian (*Burden and Hills, 1989*), and the low occurrence of *Tricolpites* sp. (*Figure 5—figure supplement 1P*) indicates a lower Albian age for the studied level (*Tanrikulu et al., 2018*).

## Resin-producing tree

Identifying which plant sources created the resin accumulations that led to the present amber deposits is still contentious, and different conifer groups have been proposed for the Cretaceous:

†Cheirolepidiaceae, Araucariaceae, and Cupressaceae in Laurasia and other groups such as †Erdtmanithecales in Gondwana (*Menor-Salván et al., 2016*; *Seyfullah et al., 2020*). The GC-MS results (*Figure 3B*, *Figure 3—figure supplement 1*) classify the Ariño amber among the amberene-rich group of Cretaceous Iberian ambers. In that regard, Ariño shows the same distribution as other Iberian ambers such as those from Peñacerrada I and San Just in terpenes **I**, **II**, **V**, **VI**, and the alkylnaphthalene **IV**, resulting from the labdane aromatisation, as well as in the overall diterpene composition (*Menor-Salván et al., 2016*). This amberene-rich group is distinguished from the abietane-rich group of Cretaceous Iberian ambers (e.g. El Soplao amber) in the lack of significant terpenes from the pimarane/abietane family as well as ferruginol, a common biomarker of extant Cupressaceae (*Menor-Salván et al., 2016*). Instead, the clerodane-family diterpene **VI** found in Ariño amber, a biomarker of the family Araucariaceae (*Cox et al., 2007*), could indicate that the botanical source of the amber is related to *Araucaria/Agathis*. In fact, the resin of *Araucaria bidwillii* is rich in kolavenic acid (*Cox et al., 2007*), which might be a biological precursor of **VI**. The Ariño amber differs from extant Araucariaceae in the lack of pimarane/abietane-class terpenoids. It is possible that early Araucariaceae lacked the biochemical routes of tricyclic diterpenoids that extant representatives possess (*Menor-Salván et al., 2016*). In any case, the most plausible stance for now is to regard the Ariño amber as resulting from araucariacean resin. The finding of charcoalified *Agathoxylon* sp. supports this interpretation, although other types of charcoalified wood likely belonging to other taxonomic groups have been found in Ariño. The presence of araucariacean remains as bioinclusions in Albian amber from the Peñacerrada I locality was proposed as evidence for an araucariacean resin-producing tree (*Kvaček et al., 2018*). Araucariaceans have also been proposed as the source of other Lower Cretaceous ambers such as those from Lebanon, Myanmar, or France (*Poinar et al., 2007*; *Perrichot et al., 2010*; *Seyfullah et al., 2018*).

## Palaeoecology

The palaeoecological reconstruction of the coastal swamp forest of Ariño that the data herein presented has allowed is remarkably complete. Floristically, the ecosystem was composed of mixed communities of gymnosperms (namely taxodioids and cheirolepids, but also araucariaceans), ferns, and angiosperms as indicated by the palynological assemblages previously obtained (*Villanueva-Amadoz et al., 2015*; *Vajda et al., 2016*) and the more diverse account presented herein, which is based on larger (or complementary in some aspects) data sampling. Based on coprolite contents, such plants were consumed by the ornithopod and nodosaurid dinosaurs described from the site (*Alcalá et al., 2012*; *Vajda et al., 2016*). As extant taxodioids are chiefly comprised of species with a high water requirement, these trees possibly were subjected to periodic flooding similarly to the bald cypress in modern swamps (*Farjon, 2005*). The extinct cheirolepids ranged from succulent, shrubby xerophytes to tall forest trees adapted to a wide range of habitats, from coasts to uplands slopes, particularly in hot and/or dry climates from lower latitudes (*Anderson et al., 2007*). Moreover, the Ariño swamp local flora was also likely encompassed by anemiacean, dicksoniacean, and/or cyatheacean ferns growing as riparian or in the understorey (*Van Konijnenburg-Van Cittert, 2002*). †Erdtmanithecales, and angiosperms, particularly those of lauralean and chloranthacean affinity, inhabited disturbed and riparian areas (*Doyle et al., 2008*). The diversity of the charophyte and ostracod fauna studied herein is higher than the previously described by *Tibert et al., 2013*, which so far can be explained based on palaeoecological constraints, notably the water salinity parameter, or sampling differences. Both charophytes and ostracods lived in shallow permanent water bodies from the freshwater swamp and were well adapted to fluctuating salinities resulting from marine inputs. The presence of the *Theriosynoecum-Cypridea-Mantelliana* ostracod association strongly evidences freshwater to slightly saline permanent water bodies (*Horne, 2009*). The intraspecific variability observed on the carapace ornamentation within the *Rosacythere denticulata* specimens is regarded as ecophenotypic (*Sames, 2011*), and could indicate an episodic increase in salinity and/or a variation of salinity, evolving towards brackish conditions.

The terrestrial arthropod community of the Ariño swamp forest was very diverse. Spiders, free-roaming or sit-and-wait lurking predators on the forest canopy or floor (*Foelix, 2011*), inhabited the palaeoecosystem, some likely using orbicular webs to hunt the abundant flying insects. The soil-dwelling arthropod fauna consisted of at least mites, jumping bristletails, cockroaches, and psocids, all of which were important for nutrient recycling (*Levings and Windsor, 1985*). The finding of a

rhagidiid mite (*Figure 4A*) is extraordinary, as the fossil record of this predatory group was limited to a few specimens in Eocene amber (*Judson and Wunderlich, 2003*). The Ariño psocid fauna differs from those previously described from other Iberian ambers, and some specimens will be described as new taxa. Extant psocids feed on algae, lichens, and fungi from diverse warm and humid habitats; such autoecology was likely already present in the group during the Cretaceous, rendering them common inhabitants of the resiniferous forests (*Álvarez-Parra et al., 2020b*). The Ariño amber insect groups with phytophagous feeding habits include thrips, hemipterans, and orthopterans. The presence of several thrips as syninclusions could suggest aggregative behaviour. The three aleyrodid hemipterans found are also preserved as syninclusions (*Figure 4I*); the extant relatives of these small sap-sucking insects mostly inhabit angiosperms (*Martin et al., 2000*), contrary to the gymnosperm affinity of the Cretaceous resiniferous trees. Fossil immature thysanopterans are rare, and the Ariño immature specimen could represent an early-stage nymph based on habitus, size, and antennal annulations and microtrichia (*Figure 4B*; *Vance, 1974*).

Holometabolous insects, overwhelmingly diverse and ecologically paramount in modern ecosystems, are well represented in Ariño amber. The exceptional discovery of a lepidopteran caterpillar, rarely encountered in Cretaceous ambers (*Haug and Haug, 2021*) and unprecedented in Iberian amber, implies herbivory not only by adults but also by immature insects in the palaeoecosystem (*Figure 4E*). The two beetle groups tentatively identified, ptinids and cantharids, have been previously found in Iberian amber (*Peris, 2020*). These, according to the habits of extant relatives, are good candidates for having engaged in trophic or even reproductive interactions with plants, as Cretaceous beetles —including some from Iberian amber— are known to have fed on pollen from both gymnosperm and angiosperms, acting as pollinators (*Peris et al., 2020*). The identified dipteran groups presently show various feeding habits, including phytophagy, mycophagy, predation, and ectoparasitism (*McAlpine et al., 1981*). Regarding the latter, female ceratopogonids likely fed on vertebrate blood, probably that from Ariño's dinosaurs according to data from other Iberian ambers (*Pérez-de la Fuente et al., 2011*). As the larval stages and adults of most of the identified dipteran groups chiefly inhabit warm and moist, often aquatic, environments such as diverse wetlands (*McAlpine et al., 1981*), these insects likely thrived in the tropical-subtropical swamp of Ariño. The genus *Burmazelmira* is currently composed of two species from younger ambers; although the discovered *Burmazelmira* sp. male (*Figure 4F*) is similar to *B. grimaldii* from San Just amber (*Arillo et al., 2018*), it shows morphological differences that could warrant describing a new taxon. Lastly, the hymenopteran groups found in Ariño amber are comprised of small to minute forms generally assumed to be idiobiont parasitoids of insect eggs. Platygastroids are the most abundant hymenopterans in Ariño amber, several of them superbly preserved (*Figure 4H*); their predominance is consistent with that observed in other Cretaceous ambers (*Ortega-Blanco et al., 2014*). One mymarommatoid specimen (*Figure 4G*) is similar to *Galloromma turolensis* (†Gallorommatidae) from San Just amber (*Ortega-Blanco et al., 2011*). The Ariño serphitids and mymarommatoids represent the oldest records worldwide for these groups. The Ariño amber has also yielded vertebrate remains, such as the oldest known mammalian hair preserved in amber (*Álvarez-Parra et al., 2020a*) and the pennaceous feather fragment reported herein (*Figure 4J*). These instances showcase the potential of this amber to provide integumentary remains of the vertebrates otherwise preserved as skeleton material in the site's rocks.

## Conclusions

Considering the extraordinary abundance and diversity of fossils that both the rocks and the amber have yielded, Ariño can be regarded as the most significant locality to date in which fossiliferous amber has been found associated with a dinosaur bonebed (*Figure 6B*). Although the amber palaeodiversity from Fouras/Bois Vert (France) could potentially match that of Ariño (*Perrichot et al., 2007*; *Tihelka et al., 2021*), the known vertebrate record from Ariño is two orders of magnitude richer, and more complete (*Néraudeau et al., 2003*). The opposite occurs in both the Pipestone Creek (Canada) and the Bone Butte (USA) localities —whereas their vertebrate/dinosaur records are at least comparable (clearly superior for Bone Butte) to those from Ariño, the palaeodiversity described as inclusions from the Ariño amber is one order of magnitude higher, with the fossiliferous potential of the amber probably being significantly greater as well (*Tanke, 2004*; *Currie et al., 2008*; *Nel et al., 2010*; *DePalma, 2010*; *Cockx et al., 2020*). Indeed, the aerial amber from Ariño stands out for being unusually highly fossiliferous, and it has already revealed a remarkable diversity in spite of the early

stages of its study, including morphotypes that will be described as new taxa. Furthermore, Ariño is the first known locality yielding fossiliferous amber and dinosaur remains in which both elements and the remaining palaeontological assemblage assessed —except some pollen and plant macroremains— generally suffered no or low-grade transport prior to burial (autochthony/parautochthony), and from which amber strictly in situ has been reported for the first time. This has enabled a reliable palaeoecological reconstruction and, more importantly, will keep allowing the extraction of sound palaeoecological inferences from upcoming material. Last but not least, Ariño is the oldest known locality preserving fossiliferous amber in a dinosaur bonebed —the only one hitherto described from the Early Cretaceous—, and it also provides the oldest fossiliferous amber from the Iberian Peninsula. All these characteristics render Ariño one of a kind, offering one of the most complete and integrated pictures from an ancient coastal ecosystem through two diverse and complementary taphonomic windows. This unique 'dual' site will remain of interest across many palaeobiological disciplines, and will be of particular significance at promoting studies in emerging fields such as deep-time arthropod-vertebrate interactions.

## Materials and methods

**Key resources table**

| Reagent type (species) or resource | Designation | Source or reference | Identifiers | Additional information |
|---|---|---|---|---|
| Chemical compound, drug | BSTFA + TMCS, 99:1 | Merck/Supelco | | |
| Chemical compound, drug | Dichloromethane Optima for HPLC and GC | Fisher Scientific | | |
| Chemical compound, drug | Methanol Optimafor HPLC and GC | Fisher Scientific | | |
| Software, algorithm | ImageFocusAlpha v. 1.3.7.12967.20180920 | Euromex | | |
| Software, algorithm | Adobe Photoshop CS6 | Adobe Systems | RRID:SCR_014199 | |
| Software, algorithm | Agilent MassHunter Quantitative Analysis B.06.00 | Agilent | RRID:SCR_015040 | |
| Software, algorithm | Microsoft Excel v. 16.0.14131.20278 | Microsoft Corporation | RRID:SCR_016137 | |

### Fieldwork and material

Amber samples were collected from the level AR-1 of the Ariño outcrop in the Santa María open-pit coal mine, near Ariño village (Teruel Province, Aragón, Spain). The amber excavation was carried out in July 2019, after two previous palaeontological amber surveys in July 2018 and May 2019 (permissions 201/10–2018 and 201/10–2019 of the Aragon Government, Spain). Excavation of aerial amber pieces was carried out at two locations from the AR-1 level (*Figure 1C*), near the AR-1/154, AR-1/156, AR-1/157, and AR-1/158 vertebrate concentrations. The acronyms of the amber pieces and bioinclusions are AR-1-A-(number). Field observations on copal associated with *Agathis australis*, herein used for comparison, were conducted at a private property in Waipapakauri, close to State Highway 1, North Island of New Zealand, by EP and XD, during a campaign in 2011 and with the permission of the landowner. Macrophotographs of the Ariño site and material were made using a Canon EOS70D.

### Amber preparation and imaging

Most of the amber pieces with bioinclusions were embedded in epoxy resin (Epo-tek 301) following *Corral et al., 1999* to facilitate their preservation and observation. Several amber pieces were cut to observe the fluid inclusions and mineralisations. The amber piece AR-1-A-2019.129 was imaged and analysed with a SEM JEOL 6010 PLUS/LA 20 kV with RX (EDS) detector at the Instituto Geológico y Minero de España laboratories (Tres Cantos, Spain). The sample AR-1-A-2018.1 of amber-infilled plant tissue was cleaved in several fragments and thin sections were made to obtain both longitudinal and transversal views of the cellular structure; other non-prepared samples were examined with a Leica Wild M3Z stereozoom microscope, equipped with a x2 frontal lens and a 0.5–40 zoom, under tangential light. Microphotographs of the amber inclusions and thin sections of amber-filled plant tissue were made with a sCMEX20 digital camera attached to an Olympus CX41 compound microscope taken through ImageFocusAlpha version 1.3.7.12967.20180920; images were processed using Photoshop CS6; fine black lines in figures indicate composition of photographs; *Figure 4C, E and F*,

*Figure 4—figure supplement 1A,G,H*, and *Figure 4—figure supplement 2C* are formed by stacking. SEM images of amber-infilled plant tissue preserving the cellular structure and charcoalified wood were obtained with a Quanta 200 electronic microscope at the Museo Nacional de Ciencias Naturales (Madrid, Spain). SEM imaging of the amber pieces with taphonomic importance was carried out with a Quanta 200 electronic microscope at the Scanning Electron Microscopy Unit of the CCiTUB (Universitat de Barcelona); all pieces, except AR-1-A-2019.79, were sputtered with graphite. The amber piece AR-1-A-2019.79 was first submerged in a 50 % solution of 37 % HCl for 2 min and then in distilled water for 1 day to remove calcium carbonate and gypsum, respectively, from the surface of the piece. The amber piece AR-1-A-2019.93 was carefully unearthed in the field, although a small protruding fragment around 3 cm long was detached. The amber piece and the small fragment were protected to avoid friction on their surface during extraction, transport, and handling. Both were submerged in distilled water for 1 day. The small fragment was treated with four ultrasonic cleaning cycles of 30 s each; it was placed in a plastic pocket bag with distilled water to avoid friction on its surface. This methodology allows an accurate visualisation of its unaltered surface at the SEM to check if it suffered abrasion.

## Amber characterisation

The FTIR (Fourier Transform Infrared Spectroscopy) analyses of the Ariño and San Just ambers were conducted using an IR PerkinElmer Frontier spectrometer that utilises a diamond ATR system with a temperature stabilised DTGS detector and a CsI beam splitter at the Molecular Spectrometry Unit of the CCiTUB. The study of molecular composition and chemotaxonomy was performed after extraction with $CH_2Cl_2$:$CH_3OH$ (DCM:MeOH 2:1) in a Soxhlet extractor using 2.3858 g of crushed stalactite-type aerial amber pieces, selected for showing the highest transparency and the least possible weathering and inclusion content. After extraction, 1.6126 g of polymeric, organic-insoluble material remained. The crude extract was directly analysed by gas chromatography-mass spectrometry (GC-MS), concentrated to 5 ml at a rotovap, and fractionated using silica gel column chromatography. Successive elution was performed using n-hexane, n-hexane:DCM 3:1 (fraction 1), DCM (fraction 2), and methanol (fraction 3). Fraction one contained the aliphatic and tetralin-rich fraction, and fraction two contained the aromatic fraction, both were analysed by GC-MS after concentration to 1 ml by evaporation in a nitrogen stream. Fraction three was dried, forming a creamy white pulverulent residue containing polar terpenoids and resin acids, analysed after conversion to trimethylsilyl derivatives by reaction with N,O-bis-(trimethylsilyl)trifluoroacetamide containing 1 % trimethylchlorosilane at 65 °C for 3 hr. GC-MS analyses were performed with an Agilent 6,850 GC coupled to an Agilent 5975 C quadrupole mass spectrometer. Separation was performed on a HP-5MS column coated with (5%-phenyl)-methylpolysiloxane (30 m long, 0.25 mm inner diameter, 0.25 μm film thickness). The operating conditions were as follows: 8 psi He carrier gas pressure, initial temperature hold at 40 °C for 1.5 min, increased from 40°C to 150°C at a rate of 15 °C/min, hold for 2 min, increased from 150°C to 255°C at a rate of 5 °C/min, held isothermal for 20 min, and finally increased to 300 °C at a rate of 5 °C/min. The sample was injected in the split mode at 50:1 with the injector temperature at 290 °C. The mass spectrometer was operated in the electron impact mode at an ionisation energy of 70 eV and scanned from 40 to 700 Da. The temperature of the ion source was 230 °C and the quadrupole temperature was 150 °C. Data were acquired and processed using the Agilent MassHunter software, and percentages were calculated by normalising the peak areas of the corresponding compounds in the total extracts. Identification of compounds was based on authentic standards and comparison of mass spectra with standard libraries and literature.

## Charophytes and ostracods

Specimens were obtained from the level AR-1 after picking the rock associated with the amber. The acronyms of the charophytes and ostracods are AR-1-CH-(number) and AR-1-OS-(number), respectively. The microfossil preparation followed standard methods in micropalaeontology as applied to charophytes (*Pérez-Cano et al., 2020*). Scanning Electron Microscope (SEM) images of selected charophyte and ostracod specimens were obtained using the Quanta 200 scanning electron microscope at the Scanning Electron Microscopy Unit of the CCiTUB. Additional SEM images of ostracods were obtained using a JEOL 6400 device at the Faculty of Earth Sciences, Geography and Astronomy, University of Vienna (Austria). Clavatoracean utricular nomenclature follows that of *Grambast, 1968*.

## Palynology

Four consecutive samples from the level AR-1 (ARN-01–ARN-04) were prepared for palynological studies by the Geologischer Dienst NRW (Germany) (https://www.gd.nrw.de). ARN-01 and ARN-02 were obtained from the lower (root) layer rich in kidney-shaped amber pieces (ARN-02 closer to the upper layer), and ARN-03 and ARN-04 were gathered from the upper (litter) layer rich in aerial amber pieces (*Figure 1B*). The rock samples were treated following standard palynological preparation techniques (*Traverse, 2007*) consisting of acid attack with HCl, HF, and diluted $HNO_3$ and sieving with different grid sizes (500, 250, 75, 50, and 12 μm). Samples were studied with an Olympus BX51 bright-field light microscope attached to a ColorView IIIu camera. The percentage ranges provided in the results show the lowest and the highest abundance of the corresponding taxon in the four samples.

## Material availability

All the material obtained prior and during the amber excavation in Ariño is housed at the Museo Aragonés de Paleontología (Fundación Conjunto Paleontológico de Teruel-Dinópolis, Teruel Province, Spain).

The copal pieces for comparison are housed at the Museo Geominero of the Instituto Geológico y Minero de España (IGME) and Universitat de Barcelona (UB).

## Acknowledgements

We are grateful to the SAMCA Group for its collaboration and to the Dirección General de Patrimonio Cultural del Gobierno de Aragón (Spain) for permissions to excavate. We thank Alejandro Gallardo (Laboratory of Palaeontology-UB), Telm Bover-Arnal (UB), Guillermo Rey, José Antonio Peñas, and the technicians of the CCiTUB. Thanks are also due to the editor and the reviewers Min Zhu and Andrew Ross, as well as an anonymous referee, for their constructive comments. Equipment has been partly funded by FEDER (IGME13-4E-1518). We are grateful to the Departamento de Ciencia, Innovación y Sociedad del Conocimiento, Gobierno de Aragón (Grupo de Investigación de Referencia E04_20 R) y Ministerio de Ciencia e Innovación, Gobierno de España (Unidad de Paleontología de Dinosaurios de Teruel and AMBERIA Team). This study is a contribution to the projects CRE CGL2017-84419, PGC2018-094034-B-C22 (both from the Ministerio de Ciencia, Innovación y Universidades, Spain, AEI/FEDER, UE), BIOGEOEVENTS CGL2015-69805-P (Ministerio de Economía y Competitividad, Spain, and the European Regional Development Fund), 2017SGR-824 (AGAUR, Generalitat de Catalunya, Spain), UNESCO IGCP Project 661 of the Austrian Academy of Sciences, and LR18 ES07 (Faculty of Sciences, Université de Tunis El Manar). JP-C acknowledges the support from the Ministerio de Ciencia e Innovación (BES-2016–076469). AS-G is funded by an APOSTD2019 Research Fellowship (Generalitat Valenciana, Spain) and the European Social Fund. RP-dlF is funded by a Museum Research Fellowship (Oxford University Museum of Natural History, UK). This study forms part of the first author's (SÁ-P) doctoral thesis, supported by a grant from the Secretaria d'Universitats i Recerca de la Generalitat de Catalunya and the European Social Fund (2020FI_B1 00002).

## Additional information

### Funding

| Funder | Grant reference number | Author |
| --- | --- | --- |
| Ministerio de Ciencia, Innovación y Universidades | CGL2017-84419 | Eduardo Barrón Xavier Delclòs |
| Ministerio de Ciencia, Innovación y Universidades | PGC2018-094034-B-C22 | Luis Alcalá |
| Ministerio de Economía y Competitividad | CGL2015-69805-P | Carles Martín-Closas |
| Generalitat de Catalunya | 2017SGR-824 | Carles Martín-Closas Xavier Delclòs |

| Funder | Grant reference number | Author |
|---|---|---|
| Generalitat de Catalunya | 2020FI_B1 00002 | Sergio Álvarez-Parra |
| Oxford University | Museum Research Fellowship | Ricardo Pérez-de la Fuente |
| Ministerio de Ciencia, Innovación y Universidades | BES-2016-076469 | Jordi Pérez-Cano |
| Austrian Academy of Sciences | Project 661 | Khaled Trabelsi |
| Université de Tunis | LR18 ES07 | Khaled Trabelsi |
| Generalitat Valenciana | APOSTD2019 | Alba Sánchez-García |
| European Regional Development Fund | IGME13-4E-1518 | Rafael P Lozano |

The funders had no role in study design, data collection and interpretation, or the decision to submit the work for publication.

## Author contributions

Sergio Álvarez-Parra, Conceptualization, Fieldwork, Investigation, Methodology, Supervision, Writing - original draft, Writing - review and editing; Ricardo Pérez-de la Fuente, Conceptualization, Fieldwork, Investigation, Writing - original draft, Writing - review and editing; Enrique Peñalver, Conceptualization, Fieldwork, Investigation, Supervision, Writing - review and editing; Eduardo Barrón, Fieldwork, Investigation, Methodology, Project administration; Luis Alcalá, Fieldwork, Investigation, Project administration; Jordi Pérez-Cano, Carles Martín-Closas, Khaled Trabelsi, César Menor-Salván, Marc Philippe, Investigation, Methodology; Nieves Meléndez, Rafael P Lozano, Fieldwork, Investigation, Methodology; Rafael López Del Valle, Fieldwork, Methodology; David Peris, Eduardo Espílez, Fieldwork, Investigation; Ana Rodrigo, Víctor Sarto i Monteys, Constanza Peña-Kairath, Fieldwork; Carlos A Bueno-Cebollada, Alba Sánchez-García, Antonio Arillo, Investigation; Luis Mampel, Investigation, Fieldwork; Xavier Delclòs, Conceptualization, Fieldwork, Investigation, Project administration, Supervision, Writing - review and editing

## Author ORCIDs

Sergio Álvarez-Parra (iD) http://orcid.org/0000-0002-0232-1647
Ricardo Pérez-de la Fuente (iD) http://orcid.org/0000-0002-2830-2639
Enrique Peñalver (iD) http://orcid.org/0000-0001-8312-6087
Eduardo Barrón (iD) http://orcid.org/0000-0003-4979-1117
Luis Alcalá (iD) http://orcid.org/0000-0002-6369-6186
Jordi Pérez-Cano (iD) http://orcid.org/0000-0002-1782-5346
Carles Martín-Closas (iD) http://orcid.org/0000-0003-4349-738X
Khaled Trabelsi (iD) http://orcid.org/0000-0003-0207-9819
Rafael López Del Valle (iD) http://orcid.org/0000-0002-7164-9558
David Peris (iD) http://orcid.org/0000-0003-4074-7400
Ana Rodrigo (iD) http://orcid.org/0000-0001-7201-9286
Víctor Sarto i Monteys (iD) http://orcid.org/0000-0003-2701-6558
Carlos A Bueno-Cebollada (iD) http://orcid.org/0000-0003-0367-4177
Marc Philippe (iD) http://orcid.org/0000-0002-4658-617X
Alba Sánchez-García (iD) http://orcid.org/0000-0003-0911-2001
Constanza Peña-Kairath (iD) http://orcid.org/0000-0002-4877-7754
Xavier Delclòs (iD) http://orcid.org/0000-0002-2233-5480

## Decision letter and Author response

Decision letter https://doi.org/10.7554/eLife.72477.sa1
Author response https://doi.org/10.7554/eLife.72477.sa2

## Additional files

### Supplementary files
• Supplementary file 1. List of palynomorphs recorded from the lower Albian bonebed level AR-1 of Ariño and their relative abundances. ARN-01 and ARN-02 were obtained from the lower root layer with kidney-shaped amber pieces, and ARN-03 and ARN-04 from the upper litter layer rich in aerial amber pieces, all of them within the level AR-1. See also *Figure 5* and *Figure 5—figure supplement 1*.

• Transparent reporting form

### Data availability
All data generated or analysed during this study are included in the manuscript and supporting files. Palynomorphs taxa and their abundances are available in Supplementary File 1. Source data of FTIR analyses are available in Figure 3-source data 1-3. Source data of GC-MS are available in Figure 3-source data 4.

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
