## [Editor Report]

In an integrative way, the authors introduced an exceptional Konservat-Lagerstätte jointly preserving dinosaur remains and fossiliferous amber. Impressively, this is the first time that strictly in situ amber is reported, and the key claims of the manuscript are well supported by the paleontological and geochemical data. This manuscript will be of broad interest to scientists, including paleontologists, geobiologists, ecologists and geologists, as well as the public.

---

## [Decision Letter]

**Decision letter after peer review:**

Thank you for submitting your article "Dinosaur bonebed amber from an original swamp forest soil" for consideration by *eLife*. Your article has been reviewed by 3 peer reviewers, including Min Zhu as Reviewing Editor and Reviewer #1, and the evaluation has been overseen by George Perry as the Senior Editor. The following individual involved in review of your submission has agreed to reveal their identity: Andrew J Ross (Reviewer #3).

Essential revisions:

1) The only significant request would be a few additional lines of text describing the outer surface of the amber pieces and any inclusions of soil in this surface (please see the notes in the attached manuscript file). If this could be paired with a single high-magnification photograph it would further support the claims made in the paper.

2) Addition of subtitles in the Discussion section will make the manuscript more readable.

*Reviewer #1:*

Álvarez-Parra and colleagues introduced an exceptional Lagerstätte jointly preserving dinosaur remains and fossiliferous amber using a multidisciplinary approach. With strictly in situ amber reported for the first time, the AR-1 Biota offers the rare opportunity to integrate data from two types of preservation (bonebed and amber), and has great potential to understand the deep-time ecological structure. The inference and the figure presentation in the manuscript are overall clear, and the conclusions are well supported by paleontological and geochemical data.

*Reviewer #2:*

The authors succeed in providing the first account of amber found in situ and in direct association with a dinosaur bonebed. The observations that they make on the physical characteristics of the amber, the chemical characteristics of the amber, and the diversity of insect inclusions will shape future interpretations of forest floor and underground amber deposits.

All of the interpretations made in the paper are thoroughly supported using the leading techniques in the field, and a balanced approach to the existing literature.

The only aspect where there might be room for improvement would be characterizing the outer surface of amber pieces formed beneath soil. The authors do an exceptional job of examining the outer surface of the amber pieces for signs of transport, but it would be valuable to know if there are soil particles included within the outer millimeters of the amber pieces, as this would strongly support their claims. If there are no soil particles within the outer layer of the amber, the claims are not necessarily undermined: this may provide a more detailed understanding of the exudation and preservational process (e.g., the resin was so viscous that soil grains only left surface impressions on the resin mass).

This paper is an exceptional study, and I recommend it for publication. There are small linguistic edits suggested throughout the manuscript. The only significant request would be a few additional lines of text describing the outer surface of the amber pieces and any inclusions of soil in this surface. If this could be paired with a single high-magnification photograph it would further support the claims made in the paper.

*Reviewer #3:*

This is a thorough study of an important new amber deposit in Spain which has preserved a variety of insects and other inclusions, some of which with further study are likely to belong to new species. The amber, which has not been transported, is found along with fossils of dinosaurs, invertebrate animals and plants, which provide a unique insight into a Cretaceous terrestrial ecosystem.

The study is a comprehensive collaboration by many experts in their fields and as such all aspects of this new and scientifically important deposit have been investigated. It represents a significant contribution to the knowledge of Cretaceous ambers and ecosystems.

This is a thorough multidisciplinary study of a new amber deposit and as such deserves to be published.

I did not find anything wrong with the paper or anything that needs to be improved.

I recommend the paper is accepted for publication.

---

## [Author Response]

Essential revisions:1) The only significant request would be a few additional lines of text describing the outer surface of the amber pieces and any inclusions of soil in this surface (please see the notes in the attached manuscript file). If this could be paired with a single high-magnification photograph it would further support the claims made in the paper.

Additional lines of text tackling this request have been added to the Discussion. The outer surface of the root amber pieces from Ariño lacks soil particles, which is related to the original inside-out growth pattern of the corresponding resin pieces, as explained below.

2) Addition of subtitles in the Discussion section will make the manuscript more readable.

Done. The Discussion has been divided into four sections, and a new Conclusions section has been added including the information present in the final part of the former Discussion.

Reviewer #2:The authors succeed in providing the first account of amber found in situ and in direct association with a dinosaur bonebed. The observations that they make on the physical characteristics of the amber, the chemical characteristics of the amber, and the diversity of insect inclusions will shape future interpretations of forest floor and underground amber deposits.All of the interpretations made in the paper are thoroughly supported using the leading techniques in the field, and a balanced approach to the existing literature.The only aspect where there might be room for improvement would be characterizing the outer surface of amber pieces formed beneath soil. The authors do an exceptional job of examining the outer surface of the amber pieces for signs of transport, but it would be valuable to know if there are soil particles included within the outer millimeters of the amber pieces, as this would strongly support their claims. If there are no soil particles within the outer layer of the amber, the claims are not necessarily undermined: this may provide a more detailed understanding of the exudation and preservational process (e.g., the resin was so viscous that soil grains only left surface impressions on the resin mass).This paper is an exceptional study, and I recommend it for publication. There are small linguistic edits suggested throughout the manuscript. The only significant request would be a few additional lines of text describing the outer surface of the amber pieces and any inclusions of soil in this surface. If this could be paired with a single high-magnification photograph it would further support the claims made in the paper.

We thank the edits made by the reviewer, as they have improved the manuscript.